# A Systematic Review of the Potential of a Dynamic Hydrogel as a Substrate for Sustainable Agriculture

**Siti Sahmsiah Sahmat** [1,2], **Mohd Y. Rafii** [1,3,*], **Yusuff Oladosu** [1], **Mashitah Jusoh** [3], **Mansor Hakiman** [3,4] **and Hasmah Mohidin** [2]

1. Institute of Tropical Agriculture and Food Security, Universiti Putra Malaysia, Serdang 43400, Selangor, Malaysia
2. Faculty of Plantation and Agrotechnology, Universiti Teknologi MARA Cawangan Sarawak, Kota Samarahan 94300, Sarawak, Malaysia
3. Department of Crop Science, Faculty of Agriculture, Universiti Putra Malaysia, Serdang 43400, Selangor, Malaysia
4. Laboratory of Bioresource Management (BIOREM), Institute of Tropical Forestry and Forest Products (INTROP), Department of Plant Protection, Faculty of Agriculture Science, Universiti Putra Malaysia, Serdang 43400, Selangor, Malaysia
* Correspondence: mrafii@upm.edu.my

**Abstract:** Adopting environmentally friendly or green technology and incorporating new alternative substrates for a sustainable agricultural industry has garnered the attention of numerous researchers. Although super absorbent hydrogels have exhibited great potential, natural hydrogel-based absorbents have gained more interest due to their environmentally safe properties. The sources for the novel green polymer are easily obtained from agricultural wastes, such as polysaccharides, agarose, chitosan, and mucilage, with zero to minimal cost. The polymer also offers several attributes, including water usage and cost efficiencies, versatile application, and increasing plant growth. Furthermore, the polymer can act as a carrier agent and aid in improving the properties of planting mediums. The present review focuses on natural and chemical hydrogel-based polymers. It discusses their potential application in sustainable agriculture and the conservation of ecosystems by providing balanced protection for seeds, plants, and soil. Future perspectives based on previous investigations are also presented.

**Keywords:** soil conditioner; super absorbance; water use efficiency; substrate; planting medium

## 1. Introduction

Ever since the last century, unprecedented and constant human population growth has been experienced worldwide. Consequently, numerous alternatives for agricultural systems were evolved to ensure that the quality and quantity of food were sufficient to meet the demand. Conventionally, crops are primarily cultivated in soils and treated with nutrients, pesticides, herbicides, and proper irrigation to meet the plant growth requirements [1]. Conventional farming techniques have several limitations that negatively impact the environment due to the chemicals' residue [2]. For example, pesticide residue could reduce the quality of agricultural goods, such as nutrient imbalance, which could then lead to serious environmental issues [3]. The ever-increasing demand for water worldwide and the consequences of climate change are putting significant pressure on the world's water resources. As a result, water scarcity is becoming a problem in arid, semi-arid, and other parts of the world. In addition, there is rivalry for the limited amount of accessible water from a variety of diverse sectors, such as urban demands, the industrial sector, and the agricultural sector, which account for more than two-thirds of the world's freshwater consumption [4]. The most significant elements influencing agricultural pro-

duction are abiotic stresses, including temperature, salinity, and drought [5]. It is anticipated that these factors will become more difficult to overcome as a result of urbanization and the deterioration of land.

Irrigation water is getting harder to come by, and people worldwide are looking for agricultural practices that maximize the amount of water used efficiently. More than 70 percent of the world's freshwater is used for agricultural purposes; however, by 2050, feeding a population of over nine billion people will require a 50 percent increase in agricultural productivity and a 15 percent increase in the demand for water [2]. On a global scale, agriculture uses more than 70 percent of the world's freshwater resources. In spite of the fact that the availability of water supply is becoming increasingly limited, agriculture continues to be one of the most prominent users of water. Irrigation can help lessen the amount of water stress in plants, which is one of the most significant factors that can have a negative impact on crop growth and output. For effective water management and continued viability, it is currently more essential than ever to conduct research on and develop novel materials. There has been a significant increase in interest in researching biodegradable hydrogels for use in commercial agricultural in recent years. This can be attributed to both the growing concern for the environment and the limited availability of water [4]. When the soil around a plant's roots begins to dry out, hydrogels as a superabsorbent polymers, can retain plant nutrients and water [5]. The volumetric water content of the soil is found to increase as a result of the gradual return of the water that was stored by the hydrogel to the soil. It also reduces the expenses of irrigation while providing plants with the necessary nutrients and moisture. Hydrogels are ideally suited for use as a secure delivery mechanism in agriculture for soil conditioners for the controlled release of fertilizers due to a number of their qualities, including slow release of water and have a high swelling capacity [1].

In agriculture, the substrates or planting medium are pivotal in supplying nutrients to the root systems [4]. Selecting planting mediums has also become critical for plant growth as seedling development is dramatically influenced by the composition of the medium [5]. Moreover, a limited amount of water in particular areas is one of the most challenging tasks for a grower to maximise profits and yield [6]. Therefore, adapting water-conserving methods becomes a potential measure to overcome the water and nutrient loss risks and combat plant moisture stress. Therefore, an improved planting medium is essential and fundamental to supporting plant growth systems and maintaining nutrients efficiently.

Extensive studies on the ability of various planting mediums, including cocopeat [4], perlites [7], rock wool [8], rice hull, bagasse, and wood residue [9], to effectively retain and deliver nutrients for plant growth have aided planters in deciding the best planting medium or substrates to cultivate crops [2,10]. Nevertheless, a basic knowledge of the properties of the selected medium is required as different mediums offer varying abilities. Excellent growing medium characteristics include porosity or sufficient aeration, water holding capacity, and cation exchange [1].

A novel soil conditioner alternative, the super absorbent hydrogel polymer, could exploit the water uptake by 60% at the initial stage of swelling [11] and maximise yield [12,13], allowing it to become a valuable option for farmers. Globally, hydrogel application as a planting medium has captured interest in the academic and industrial sectors. Furthermore, the substance has been applied in several technologies, such as engineering [14], medical and health sciences [13–16], and agriculture [6,17,18].

Hydrogel has been defined in various ways, such as water absorbing material, water superabsorbent, and crosslinked polymer, indicating its ability to retain water within its structure without losing its structure in the water [19]. Among the countries that have actively researched hydrogel applications in agriculture and horticulture as a soil conditioner and controlled release device within 2018–2022 include Brazil researched on watermelon in the sand and vermiculite and hydrogel's development from the cashew tree gum

as controlled released device [13,20], China worked on activated-carbon-filled agarose hydrogel for seed germination [21,22], Egypt studied on guava, wheat, faba bean and on the sandy soil [12,23–25], India working on blackgram, ornamental plant and marigold in the loam, clay, sand and compost mixture [4,26,27], Iran explored on cucumber and olive under the limitation of water [7,28], and Italy that researched on sweet basil, Mediterranean cultivars and bean in red and white soil [29–31].

Scholars have been investigating the effects of superabsorbent hydrogels as a soil amendment on various soil types (sand, sandy loam, peat) [26,32,33], dosage rates [11,29,34], and base types [18,33,35] on plant growth [4,36]. Nevertheless, although numerous studies focused on applying hydrogel in soilless agriculture, systematic reviews of the existing reports are still limited. Moreover, a conventional review could lead to reviewers' biases in evaluating the literature and rarely considers the differences in the quality of studies without comprehensively covering the topic reviewed [37].

A systematic review is an organised and planned method describing rationale and hypothesis [38,39] with a defined research question [39]. The technique involves quantitative and qualitative identifications and evaluations of the data in the searched review by critically appraising and judging previous studies by employing explicit and transparent approaches to answer the research questions [40,41]. Besides, systematic reviews provide a reliable and comprehensive judgment on the findings, detail the review process, and are reproducible [40].

Systematic reviews of identifying and developing potential hydrogel applications as a sustainable agricultural substrate remain limited, although a vast body of literature on the relevance of hydrogel in agriculture is available [15,42–45]. The current analysis provided the capacities of applying hydrogel as an agricultural substrate with systematic and in-depth information on the review procedures, including the keywords, identification, article screening, article eligibility, and database. The present study provides a platform to assist future research by proctoring knowledge and understanding of the types of hydrogels and their potential applications in agriculture. The superior hydrogel properties from previous literature are elicited in this study, offering a systematic review of the relevance of hydrogel implementation within the years 2018–2022. The study was guided by the main research question 'What are the potential applications of hydrogels as substrates for sustainable agriculture?'.

## 2. Methodology

The methodology adopted in the present systematic literature review (SLR) was divided into three sub-sections, namely database resources, systematic searching strategy, and data analysis, and was guided by the Preferred Reporting Items for Systematic Reviews and Meta-Analyses (PRISMA) [39]. First, the research question was formulated by applying the population, interventions or interest, comparators and outcomes, or PICo approach, which in this study corresponded to soilless agriculture, hydrogel, and substrates or planting medium, respectively [45]. The primary research question was "What is the potential application of hydrogel in agriculture?" while the sub-research questions were "What are the benefits of using hydrogel in a soilless system?" and "What are the limitations of hydrogel in the soilless system for sustainable crop production?".

### 2.1. Review Protocol-PRISMA

The present SLR was guided by PRISMA [39], which provided the review with organisation, evaluation, and examination to ensure its quality and transparency [46]. Although the PRISMA protocol is commonly employed in medical studies, the procedure has directed several researchers in the agriculture field to conduct SLR with clearly defined research questions successfully [47,48].

*2.2. Database Resources*

Two primary databases were utilised in the present study: the Web of Sciences (WoS) and Scopus. The WoS recorded a robust database that covered 8372 journals with 200 categories related to the subject, including agronomy and horticulture. The Scopus indexes comprised diverse subject areas, such as agricultural and biological sciences, environmental science, biochemistry, genetics, and molecular biology, and consisted of 2940 journals associated with hydrogel in agriculture. Nevertheless, it was worth noting that no databases were comprehensive enough to cover all relevant topics in this study. Consequently, additional specialised and recognised databases, including ScienceDirect (987), Springer (1377), and CABI (997), were considered to enhance the possibilities of procuring relevant and adequate publications for this SLR [49,50].

*2.3. Search Strategy*

The selection of relevant articles for the current study was performed systematically, including the identification, screening, and eligibility, as proposed by Shaffril et al. [51]. The processes are explained in the subsequent sub-sections.

2.3.1. Search Strings Identification

The first stage in the identification process was the search string concerning the topic and research question. Table 1 lists the search string employed to search the primary databases, which comprised suitable pre-determined keywords. The search strings hydrogel, planting medium, soilless, fertigation, agriculture, and cultivation resulted in 380 article retrievals in the first stage of the systematic review process. A total of 129 journals were retrieved from WoS and Scopus, while manual searches with similar keywords in other databases yielded 25 (ScienceDirect), 121 (Springer), and 106 (CABI) articles.

**Table 1.** The search strings employed in the SLR.

| Database | Search Strings |
|---|---|
| Web of Sciences (WoS) | (((hydrogel*) AND (plant*) AND (med*) AND (soil* OR fertigation*) AND (agr* OR cultiv*))) |
| Scopus | TITLE-ABS-KEY (((hydrogel*) AND (plant*) AND (med*) AND (soil* OR fertigation*) AND (agr* OR cultiv*))) |

2.3.2. Screening

The second stage of this review was the screening stage, during which inclusion and exclusion were conducted on 353 articles. Based on the pre-determined criteria, 28 articles were excluded. The first characteristic considered was the type of publication or document. The present study focused on articles with empirical data as the primary sources to ensure the quality of the review, which eliminated any publications from studies, conference papers, book chapters, editorials, short surveys, and notes. Non-English entries were also excluded to avoid confusion and translating challenges. Moreover, only articles published between 2018–2022 were included in the review. Consequently, 270 articles were eliminated from the list based on the criteria shown in Table 2 post the first screening.

**Table 2.** The inclusion and exclusion criteria of the SLR.

| Criteria | Inclusion | Exclusion |
|---|---|---|
| Document type | Articles with empirical data | Reviews, conference papers, book chapters, editorials, short surveys, and notes |
| Language | English | Non-English |
| Timeline | 2018–2022 | <2018 |

### 2.3.3. Eligibility

The third stage in the SLR was eligibility evaluation. The titles, abstracts, and main contents were manually perused to ensure that the criteria of the articles were sufficient and met the objectives of the studies. The process resulted in the deletion of 25 additional articles that emphasised the production of hydrogel and were review publications as this SLR focused on empirical data and the potential of hydrogels as planting mediums. Resultantly, only 58 articles were qualitatively synthesised in the current study. Figure 1 illustrates the overall process involved in the systematic searching strategy opted for the present SLR.

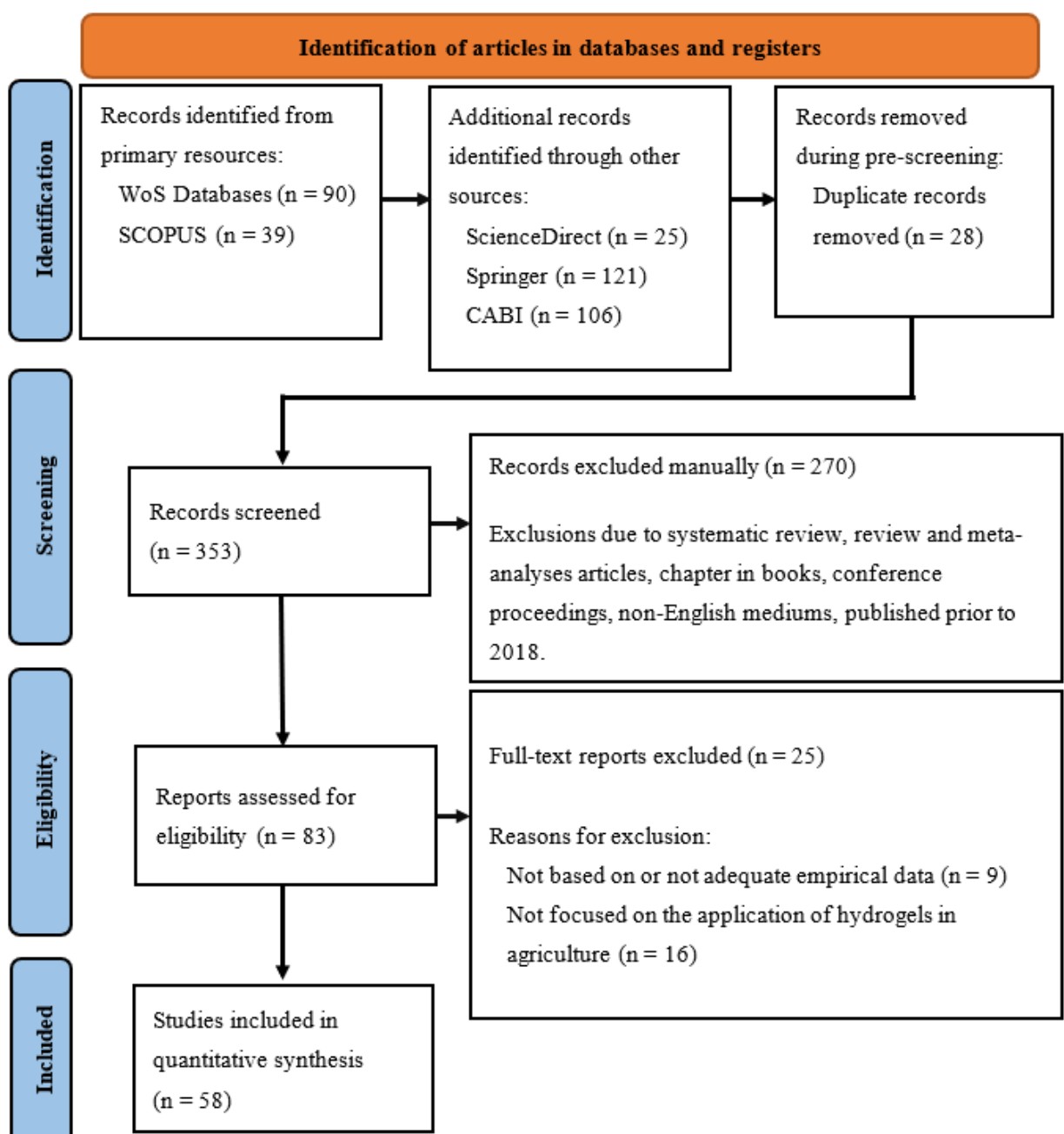

**Figure 1.** The PRISMA flow diagram of the study *. * Note: Adapted from [39].

2.4. Data Abstraction and Analysis

After the screening and eligibility stages, the articles were evaluated, reviewed, and analysed in correspondence to the research question using a quantitative or qualitative synthesis that allow researchers to compare the data sources for a review analysis [40,52]. Subsequently, the themes and sub-themes were developed by compiling and scrutinising the information based on the thematic assessment of the 58 screened articles. The first steps in the data abstraction were compiling all relevant statements that met the research questions pooled in directive groups and demonstrating connections between raw data.

The raw data were converted into utilisable data through theme identification, concept, or related ideas. The six main themes developed were hydrogel, planting medium, effects of hydrogel on planting medium properties, potential as planting media and as a carrier, the physical properties of the hydrogel, adverse impacts and future perspectives. Finally, 48 articles were manually added to complement the designed themes, which addressed the potential of hydrogel applications in agriculture.

## 3. Results and Discussion

### 3.1. Hydrogel

This review employs the terms superabsorbent, polymer, and hydrogels interchangeably. Superabsorbent hydrogel or polymer is a generic term referring to the three-dimensional (3D) matrixes utilised in medical, engineering, wastewater treatment, food processing, and agriculture [13,53]. The ability of hydrogel polymers to retain water has captured the interest of researchers, hence prompting the investigation into its potential in agriculture.

3.1.1. Types of Hydrogel Bases

Hydrogel-based sources could be categorised as synthetic- and natural-based and hybrid [29]. A majority of commercial hydrogels are obtained from synthetic polymers (polycaprolactone, polyvinyl pyrrolidone, polylactic acid and polyvinyl alcohol) prepared through different methods, such as polymerisation [20], ionic interaction, hydrogen or physical bonding, and hydrophobic interactions in 3D network connections that form giant molecules [23].

1.　Synthetic Hydrogels

Commonly, the polymers employed to produce synthetic hydrogels consist of polycaprolactone, polyvinyl alcohol, pyrrolidone, acids (citric, tartaric, and lactic acids), and polyethylene glycols acrylamide for crosslinking purposes [54–56]. Synthetic polymers possess a unique ability to retain more water with better durability [57]. Nonetheless, a major issue with this hydrogel is its disposal and non-renewability features that contribute to landfill contaminations and biodegradability challenges [55]. Moreover, the polymerisation of polyacrylic acid has become a significant contaminant concern that severely threatens the health of humans and microflora [19,58]. Consequently, other hydrogel preparation blends have mitigated the biodegradability issue by improving the physicochemical and mechanical features [56], which is more attractive for agriculture.

2.　Natural-Based Hydrogels

The selection of polymers to manufacture hydrogels has shifted from synthetic to more environmentally friendly and non-toxic materials and offers low-cost production and high hydrophilic features [17]. Among the natural polymers commonly utilised to produce hydrogels are gelatin [25,59,60] and polysaccharides, such as alginate [60–64], agarose [21,54] and chitosan [4,43,56].

Mucilage is an example of natural hydrogel with excellent surface tension, contact angle, and viscosity [65]. The substance reduces water flow under dry conditions and manipulates pore size distribution into a narrower range [53]. Surface tension is closely related to water holding capacities, which is beneficial for seeds treated with mucilage to

germinate at low water potential, high temperatures or under stressful conditions [36]. Furthermore, a mucilage layer could be an oxygen barrier and slow seed germination [36], providing a mechanism for seeds to stay longer in soil seed banks for later germination under more favourable conditions. Alternatively, hydrogel application has been reported to delay the wilting point of seedlings by ten days, reaching 1500 kPa matric potential [65], and improve the stress tolerance of plants [65].

Another natural-based hydrogel employed as planting mediums amendment is chitosan [43]. Chitosan-based hydrogels are eco-friendly soil conditioners that were reported to increase water holding capacity by up to 154% and enhance nitrogen (N) contents in planting mediums by double compared to the control treatment [43]. The plant showed a high capacity of ammonium ion ($NH_4^+$), when N adsorption was planted in the hydrogel-grafted chitosan and it maintained the availability of the plant ($NH_4^+$)—N concentration in the soil [17]. Consequently, chitosan hydrogel applications could be considered as a N slow-release device mechanism that would maintain available $NH_4^+$ and N concentration levels in soils [17].

Alginate is a polysaccharide-based hydrogel with unique characteristics due to its thickening and stability while efficiently binding with other divalent metal ions [11]. The application of alginate is not limited to agriculture as a nutrient carrier but applies to food thickeners, additives, gelling agents, and green packaging materials [55]. A study revealed that the fertilisers in planting mediums were efficiently released [66] and prolonged in alginate polymer during a forty-five-day application [11]. Nonetheless, when applied on its own in entomopathogenic applications, alginate exhibited low stability and a short life span prone to desiccation, leading to isomerisation changes and it was also time-consuming and expensive [67]. Consequently, alginate hydrogel is typically grafted or augmented with another excipient, such as inulin, trehalose, or chitosan, to overcome the shortcomings before its utilisation in agriculture [68].

### 3.1.2. Hydrogel Characteristics

Numerous literatures have cited reasons for adopting hydrogel in agricultural settings as a planting medium. Most reports discussed the positives, biodegradability, phytotoxicity, and plant growth performances [29,35]. Moreover, the awareness to protect the environment has improved, which led to the intended application of biodegradable materials as alternatives in any agricultural implementation [69], including hydrogel production [44]. An advantage of employing biodegradable materials includes reduced production costs as the substances are commonly derived from waste resources, such as lignin [23,37], chitosan [44,68], and cellulose [70]. Several reports have also demonstrated that biodegradable hydrogels possessed better absorbency and strength [30] than the synthetic polymer due to their homogenous microporous structure, which is beneficial in soilless systems and seed germinations, in which its structure allows sufficient diffusion of oxygen functional groups. Conversely, the swelling capacity of biodegradable lignin-based hydrogel was recorded to be lower than commercial synthetic hydrogels; however, it documented faster deterioration rates after a year of application without phytotoxicity symptoms [35].

### 3.2. The Effects of Hydrogels on Planting Medium Properties

### 3.2.1. Physical Properties

Among the characteristics of an excellent growing medium are porosity [71], water holding capacity [6,42] and cation exchange abilities [28]. The conventional approach to growing plants requires soil. Nonetheless, soils typically possess poor aeration, drainage, and water-holding abilities and could harbour numerous microorganisms and debris on the surfaces [72]. Consequently, utilising another alternative as a growing media has captured global interest as it might reduce the risk of soil-borne diseases and diminish labour requirements due to less irrigation frequency to tackle eradication process challenges [7].

1. Bulk Density and Water Holding Capacity

Bulk density is a critical physical property as it could influence soil's porosity, moisture, and hydraulic conductivity of [73]. Moreover, a high bulk density value determines the compactness of the soil, hence affecting air and water movements, which could impede root growth and yield [74]. Accordingly, improving the planting medium could allow for superior gas exchange and sufficient water and nutrient supply through efficient rooting systems and mechanical support for plant growth [75,76].

The addition of hydrogels significantly influenced the physiological and chemical attributes of soil, where its bulk density was diminished by 60% and water holding capacity (WHC) to the substrates was enhanced [37,43,77,78], resulting in a higher total pore space for roots intrusion [73]. The hydrogel-based method is also utilisable in vivo root phenotyping to prove the interconnected pore space filled by the nutrient-held hydrogel [76]. As WHC directly affects the percentage of pore space [73–78], the polymer could reduce irrigation requirements [25,31,73] and mitigate water limitations in particular areas. Furthermore, when considering fertiliser costs, adding hydrogel might be the ideal option to optimise water and nutrient employment while maintaining crop production [31,70,79].

The capabilities of hydrogels to reduce irrigation requirements while increasing production yield, have captured the interest of growers to include the polymer in their costing [1,25,29]. Numerous literature cited the advantages of adding hydrogel in planting media, which resulted in tremendous increments in porosity, hence providing more oxygenation to plant systems for better performances [33,80]. Moreover, amendment with the polymer reduced irrigation requirements [70,78] without harmful effects on the quality of the crops [7].

2. Soil Water Content at Field Capacity

The changes in field capacity with hydrogel application aided soil moisture retention and plant growth [34]. Good soil water content allows water withdrawal from the soil into the root system and generates turgor pressure in plant cells [81]. The ability of hydrogels to augment water retention for a longer time in the rhizosphere played an essential role in root development and seed germination by facilitating the translocation of water uptake and dissolving nutrients and air in soil [29]. Moreover, hydrogel polymer applied as a water retention amendment in sloping topographies for field trials documented a 3 to 5% yield increase [70].

3. Soil Water Content at Permanent Wilting Point

The rapid diffusion of water through plant systems occurs when the soil moisture borders field capacity. In an investigation, the amended hydrogel planting medium demonstrated improved soil water content [33] and physicochemical properties [7] and delayed wilting point, depending on the soil types and temperature. Furthermore, the higher amount of hydrogel applied resulted in it being more efficient as a water reservoir due to its high water absorbency [30,43]. Nevertheless, as the planting medium approaches the permanent wilting point, which was estimated at −1500 kPa matric, the effects of the hydrogel to retain water decreased [65], indicating that the water retained by the hydrogel might not be available for the plant when necessary [12].

3.2.2. Biological Properties

Living organisms, such as earthworms and bacteria, are essential indicators of healthy soil [82,83]. The larger organisms would create pores resulted an aeration in the ground while smaller organisms, such as bacteria, fungi, yeast, and algae, aid in decomposing organic matter [84]. The decayed organic matter plays an essential role in nutrient cycling, improving soil structure by providing more nutrients to soil organisms, increasing its water holding capacity, and reducing the risk of erosion and leaching [85].

Applying hydrogel as an amendment to planting mediums in proper concentrations would not alter the soil microbiocenoses [19]. The biodegradation study by vermicompost

showed that the polymer could be turned into compost, further confirmed by healthier cotton seed germination placed on the compost [55]. In addition, the activities of the effective microorganisms in hydrogel-loading medium are still capable to increase the nutrient content, reducing the soil pH, and increasing the organic matter [24]. Furthermore, several investigations on the particle size of growth mediums proved a significantly enhanced root fresh weight and length [86] and water retention characteristics [87]. Nonetheless, various organic amendments have been reported to affect soil N mineralisation [88], thus requiring consideration when selecting fertilisation management.

### 3.2.3. Chemical Properties

Plants grown on soilless media are required to possess the capabilities of absorbing similar nutrients to their soil counterparts [89]. The growth medium's pH directly affects the plants' ability to absorb the available nutrients [73]. Some species' tolerance toward higher or lower pH level varies; however, optimal plant growth usually requires a pH between 5.5 and 6.5. Typically, nutrients are negatively charged ions and are attracted to positively charged ions, such as $NH_4^+$, potassium ($K^+$), calcium ($Ca^{2+}$), and magnesium ($Mg^{2+}$) [17]. The ability of the cultivation medium to adsorb the cations is known as cation exchange capacity (CEC) which could be manipulated through soil amendment, such as fertiliser application, to improve soil properties during low nutrient provision conditions [32]. In soilless agriculture, controlling the number of nutrients and pH in the cultivation medium is easily performed by regular monitoring via a portable conductivity meter and a pH meter [33,90].

The application of hydrogel amendments in planting mediums would not disturb soil activities, including carbon mineralisation and carbon dioxide ($CO_2$) emissions to the atmosphere [19]. In reality, $NH_4^+$ and N are lost through volatilisation, nitrification, and denitrification, which result in poor soil quality that negatively affects plant growth [17]. Hydrogels were reported to diminish the $NH_4^+$ and N in the soil and mitigate the soil microbiota ecosystem, nitrification, and denitrification, thereby delaying denitrification by reducing the risk of N loss from the ground in a short time [17].

### 3.3. Potential as Planting Media

### 3.3.1. Plant Growth Performances

Several studies on superabsorbent polymer hydrogel as a planting medium demonstrated positive effects on plant growth performances and physiological traits, *viz.* improved turf index quality, colour, density, ground cover, fruits, and essential oil yield in sweet basil and lemon [78] without compromising their physiological traits [31–33,91].

The physicochemical properties of zeolite amended hydrogel substrate was reported to possess enhanced water retention and improved plant growth parameters, including height, leaf number, buds, flower, branches, and nodes [28,80], while wheat recorded faster germination rate, biomass, and protein yield [92]. Furthermore, a suitable dose or concentration of hydrogel applied in a planting media [83], which aided in water availability manipulation, significantly affected the physiological parameters of the plants cultivated [73,93,94]. The application of hydrogel in a planting media that alleviates water stress by increasing the stability of the cell membrane and leaf relative water content, as well as reduce the risk of blockage of xylem and phloem for translocation [95]. Hence, the medium can retain available water longer for plant used [1] and improv physiological parameter.

### 3.3.2. Water Retention Capacities

The hydrophilic attributes of the super absorbance hydrogel contributed to its capacity to store water approximately 200–1000 times its original weight [7,13]. Studies also reported that the polymer has significantly affected plant growth when applied in areas with water scarcity and arid and semi-arid regions [81,94,95]. Moreover, amendment with the polymer resulted in increased soil saturation, less nutrient leaching in irrigation, and higher nutrient uptake by the plants [26,94]. The application also recorded a significant reduction in irrigation requirements with water-saving between 170–300% [78,96]. Consequently, numerous growers tend to adopt the hydrogel application, including gerbera [1,73], paddy [94], *Euonymus japonicus* 'Aureomarginatus [18], and local beans in Southern Italy [31].

### 3.4. Potential as Carriers

The robust growth in global industrial and agricultural sectors has led to the excessive generation and utilisation of chemicals that could result in a catastrophe if employed inappropriately in the environment, such as nutrient imbalance and reduced quality of agricultural products [97]. Worldwide pesticide utilisation is estimated at two million tonnes (47.5% herbicides, 29.5% insecticides, 17.5% fungicides, and 5.5% other pesticides), with China, the United States of America (USA), and Argentina as the leading countries [98]. Accordingly, developing hydrogels that could be dissolved, dispersed, or encapsulated offers an alternative for preventing crop loss due to pests and diseases and a solution to mitigate the adverse effects of direct pesticide application [43].

### 3.4.1. Insecticides and Fungicides

The potential of hydrogels as insecticide carriers, as demonstrated by alginate hydrogels, could aid against pests [99]. For instance, the application was beneficial against plant root-feeding insects [67], reduced *Phytophthora capsici* infestation in capsicum [56], and controlled ant population in trees within 48 h of application [99]. Nonetheless, incorporating active insecticidal ingredients into hydrogel requires further discussion as it has a different mode of action against pests [1,67]. The optimal dosage applied in the hydrogel that could provide maximum protection also necessitates determination to prevent wastage and adverse environmental effects [33].

### 3.4.2. Fertilisers Hydrogel

The practical application of hydrogel in several fields has received increasing attention and led to the development of fertiliser-integrated hydrogels [100]. Studies also suggested that hydrogels exhibited significant effects as carriers for fertilisers without hindering crop performance [94] and nutritional values. Furthermore, intelligent and innovative fertiliser-hydrogels encapsulated with N, phosphorus (P), and K (NPK) [101] could also amend the poor compound-release biodegradation mechanism kinetics in agricultural soils. Nonetheless, varying hydrogels demonstrated different water-releasing mechanisms when applied to different soil types [30] due to the varied polymer bases employed for graft during production [11]. Accordingly, selecting the accurate hydrogel base that matches the soil is vital to maximising the potential of the polymer [34,37,80].

Hydrogels incorporated with fertilisers recorded altered swelling capacity, influencing their crystallinity, morphology, hardness, and adhesiveness and could be recycled for over 55 consecutive cycles [20], hence permitting labour and fertiliser cost efficiencies. A slow-release hydrogel incorporated with urea fertiliser was also reported to improve N supply and plant growth while minimising environmental hazards [102] and P and K accumulations compared to other fertilisers [103]. In another investigation, a synthetic biopolymer was proven a good companion when paired with beneficial microorganisms, which created a mutual symbiotic relationship with root systems that aided in P and K exchanges, and N fixation and increased organic matter mineralisation rates [24]. The bio-

fertiliser polymer also promoted plant growth and enhanced nutrient uptake by alleviating abiotic pressures, such as water and salinity stresses [24].

3.5. Negative Effects

Reports on the adverse impacts of hydrogel applications as a planting medium are lacking as most researchers addressed the water retention abilities of hydrogels. The poly-electrolytes employed in hydrogel manufacturing commonly contain acrylamide, acrylic acid, or potassium acrylate with negative charges [8,18,95,102]. Minimising or neutralising the acrylic acid converted into dimers at the end of the polymerization process is necessary to prevent adverse effects on plant performance and achieve the maximum potential of the polymers in aiding plant growth [24].

The main concern of employing chemical cross-linker during hydrogel crosslinking is their toxicity [24]. Initiating the polymerisation of acrylic acid bearing a carboxyl group, such as -OH, -CONH, -CONH$_2$, and -COOH [24], contribute to the water-sorbing properties of the hydrogel, which is pronounced in high salinity areas [103,104]. The addition of sodium hydroxide (NaOH) or potassium hydroxide (KOH) to encourage the dissolution and induce the gelatinisation of the polymer or neutralise the acrylic acid leads to the accumulation of Na and K in high concentrations [103], resulting in salinity toxicity and inhibiting plant growth [6]. Furthermore, calcium (Ca) is poorly absorbed by the plants planted in the geocomposite polymers [104]. The plants cultivated in the growth medium amended with the hydrogel recorded development inconsistencies and Ca deficiency in their young leaves, withered roots due to excessive sodium accumulation [73], plant biomass reduction, and some base tips [13].

The osmotic difference between the media and the hydrogel created a concentration slope that led to water diffusing from the polymer, which was replaced by solutes [104]. The high osmotic pressure could harm the plant and negatively affect the water-absorbing power of its root system [73]. The high solute concentration in the solution also reduced swelling capacity and resulted in plant stress [35], while the excess Na uptake by the plant roots could explain the impairments in the plant cell membranes [73]. Consequently, the potential of hydrogels as a solution to absorb xenobiotic compounds in soils should be explored to reduce the greenhouse effect risks.

*3.6. Future Perspectives*

The findings of this systematic review led to several recommendations that might be beneficial in future research. Firstly, producing a more environmentally safe hydrogel that decomposes rapidly in the soil must be considered. In 2019, the agricultural sector contributed 7.1% (RM101.5 billion) to the Gross Domestic Product (GDP), in which oil palm was the primary contributor (37.7%), followed by other agriculture (25.9%), livestock (15.9%), fishing (12%), forestry and logging (6.3%), and rubber (3%) as reported by the Department of Statistic Malaysia (DOSM) (2020). Unfortunately, agricultural wastes are often discarded as they are costly to handle, process, and store [105]. Nevertheless, biomass wastes comprise cellulose, a valuable hydrogel source. The compound is also biodegradable and environmentally safe [20,101]. Moreover, cellulose could be employed as a substitute for acrylic acid in hydrogel manufacturing to solve the toxicity issue [30].

Most of the studies reviewed in the current SLR focused on southern and northern Asia, the Middle East, and European countries. Nevertheless, no academic reports were published in the Southeast Asia regions (e.g., Malaysia, Thailand, Indonesia, and Vietnam) between 2018–2022 on the selected database, demonstrating the need to conduct similar application research in the areas. Southeast Asia countries might benefit from manufacturing biodegradable hydrogels as they are among the most significant countries producing agricultural waste from their primary economic activity, which was supported by the report in the Association of South East Asian Nations (ASEAN) Statistical Yearbook [106].

## 4. Conclusions

On a global scale, innovative new approaches to plant protection are currently being researched and developed to reduce reliance on synthetic and chemical pesticides, save the environment, and safeguard biological species. Taking this into consideration, polysaccharide hydrogels have been produced and used in agriculture for various purposes (slow pesticide release, seed coating, soil conditioners, and fertilizer carriers). Superabsorbent hydrogel polymers possess promising potential in agriculture evolution. The present review attempted to update and discuss hydrogel application capacities in agriculture as planting media and carriers and its effects on the physical properties of soil. The superabsorbent hydrogel could be derived from plant-based waste materials, which are naturally biodegradable, economically viable, and possess versatile utilisations. Moreover, the yield of some crops recorded a significant increase over the conventional method, indicating a possible agronomic practice as a planting medium. The application of the hydrogel improved the physical properties of the medium and was biota friendly. Furthermore, the employment of the hydrogel would benefit in several aspects as it significantly reduced irrigation requirements in planting systems. The advantages of utilising the polymer outweigh the disadvantages and thus could be considered a novel substrate for sustainable agriculture. Consequently, future considerations should include improving the formulation for hydrogel production with more super absorbance properties and the ability to be employed in more production cycles.

**Author Contributions:** S.S.S. and M.Y.R. drafted the original manuscript, while the editing, finishing, and proofreading were carried out by S.S.S., M.Y.R., Y.O., M.J., M.H. and H.M. All named authors contributed to the work by offering recommendations on the original draft manuscript. The paper has been reviewed and approved by all of the authors in its published form. All authors have read and agreed to the published version of the manuscript.

**Funding:** This research was sponsored by a Long-Term Research Grant Scheme (LRGS/1/2019/UKM/01/5/4) with Vot number 6300242 from the Malaysian Ministry of Higher Education for food security and sustainable vegetable production technology in urban agriculture.

**Institutional Review Board Statement:** Not applicable.

**Informed Consent Statement:** Not applicable.

**Data Availability Statement:** All data to support the finding in this manuscript is presented within.

**Acknowledgments:** The authors also thank the Institute of Tropical Agriculture and Food Security, Universiti Putra Malaysia, for the assistance provided and Universiti Teknologi MARA (UiTM) for granting a study fellowship to the first author.

**Conflicts of Interest:** The authors declare no conflict of interest.

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
