# Peer review of "A Systematic Review of the Potential of a Dynamic Hydrogel as a Substrate for Sustainable Agriculture"

_horticulturae, doi:10.3390/horticulturae8111026_

Round 1

Reviewer 1 Report

Dear authors, 

Your systematic review is well organized and performed.

There are still need to revise some parts as follows:

1) Line 70: Is it necessary to separate subsection 1.1. as there are no other subsections?

2) Line 128: Please, add the authors name for this reference [52] - " ... as proposed by Shaffril et al. [52]"

3) Lines 252-253: Something is wrong here - "Plant roots in chitosan hydrogel amended soil also absorbed and adsorbed high amounts of ammonium ions .."

4) Line 382: Do you think that "capacity" is more appropriate than "ability"?

5) Line 732, 742, 760: The Latin name of species should be in italic

6) Please, correct the formula of "NH4 +" where needed in the text (4+ should be a superscript)

Reviewer 2 Report

it is a good work, well organized, you should pay attention to the suggestions and recommendations in the pdf manuscript.

Reviewer 3 Report

See my comments on the attached file, please

Reviewer 4 Report

Dear Authors,

The review paper presents interesting and importante knowledge regarding the dynamic hydrogel  application in agriculture. The paper is overall well written, organized and the literature updated.

Still, some corrections need to be done and they are marked on the manuscript itself.

Best Regards,
